# Explaining the Plans of Agents via Theory of Mind

**Maayan Shvo[†], Toryn Q. Klassen, Sheila A. McIlraith[†]**

Department of Computer Science, University of Toronto, Toronto, Canada
Vector Institute, Toronto, Canada
[†] Schwartz Reisman Institute for Technology and Society, Toronto, Canada
{maayanshvo,toryn,sheila}@cs.toronto.edu

## Abstract

For a plan to achieve some goal – to be valid – a set of sufficient and necessary conditions must hold. In dynamic settings, agents may come to hold false beliefs about these conditions and, by extension, about the validity of their plans or the plans of other agents. Since different agents often believe different things about the world and about the beliefs of other agents, discrepancies may occur between agents' beliefs about the validity of plans. In this work, we explore how agents can use their Theory in Mind to identify and correct other agents' beliefs that give rise to discrepancies pertaining to plan validity. We appeal to an epistemic logic framework to allow agents to reason over the nested beliefs of other agents. We realize our approach using epistemic planning and demonstrate how an off-the-shelf epistemic planner can be used to resolve discrepancies regarding plan validity in a number of domains. A study showcases our approach's ability to resolve misconceptions held by humans pertaining to plan validity.

## 1 Introduction

*"Planning is the art of thinking before acting"* (Haslum 2014), but a problem with thinking before acting is that the validity of the resultant plan is predicated on *beliefs* about the way the world is, rather than ground truth, and even if those beliefs are correct at the time of planning (and they may not be!), the actual state of the world may change prior to plan execution, invalidating the plan, sometimes unbeknownst to various agents. Moreover, agents may perceive discrepancies between their own beliefs and other agents' beliefs about the validity of plans (e.g., Alice believes that Bob's plan is not valid but that *he* believes it is). Here we wish to allow agents to contemplate each others' plans, realize when agents hold misconceptions about the validity of their plans or the plans of other agents and communicate with those agents to repair believed misconceptions.

To contemplate another agent's beliefs and plans, agents must employ their Theory of Mind which, according to Premack and Woodruff (1978), is exercised when an agent imputes mental states (e.g., plans, goals, and beliefs) to itself and others. To enable agents to employ their Theory of Mind, we appeal to epistemic logic and propose a framework that allows an agent to identify and resolve discrepancies between their beliefs and the beliefs of other agents regarding plan validity.

A large body of previous work emphasized the role of Theory of Mind in resolving discrepancies between agents about various properties of plans (e.g., optimality and validity). For instance, the body of work on *model reconciliation* has extensively investigated how to enable agents to use their Theory of Mind and resolve discrepancies between agents in the context of plan explanation (Chakraborti et al. 2017; Sreedharan, Chakraborti, and Kambhampati 2021). In this paper we broaden the discussion of the role of Theory of Mind in such settings. In particular, the expressive nature of our framework allows agents to identify and resolve misconceptions held by agents about the beliefs of other agents about plan validity (e.g., in order to resolve Mary's misconception about Bob's beliefs about the validity of his plan, Alice may inform Mary that Bob holds a false belief about some fact relevant to the plan's success). Moreover, our framework allows agents to correct beliefs that threaten the achievement of *epistemic goals* (where an agent is trying to achieve some state of knowledge or belief).

To realize our approach, we establish a relationship between our proposed formulation for discrepancy resolution and *epistemic planning* which, as a field, is focused on generating plans to achieve epistemic goals in the context of the beliefs and knowledge of agents (e.g., Petrick and Bacchus, 2002; Bolander and Andersen, 2011; Kominis and Geffner, 2015; Muise et al., 2015; Huang et al., 2017; Le et al., 2018; Fabiano et al., 2020). We demonstrate the feasibility of resolving discrepancies via epistemic planning by evaluating our approach on a diversity of domains, using an off-the-shelf epistemic planner. Finally, we conduct a study which indicates that our approach can effectively resolve misconceptions held by humans pertaining to plan validity.

## 2 Preliminaries

**KD45$_n$.** We briefly discuss the multi-agent modal logic KD45$_n$ which we appeal to in this work (Fagin et al. 2004). Let $Ag$ and $\mathcal{P}$ be finite sets of agents and atoms, respectively. $\phi$ and $\psi$ are used to represent formulae and $\top$ and $\bot$ to represent *true* and *false*, respectively. The language $\mathcal{L}$ of multi-agent modal logic is generated by the following BNF:

$$\phi ::= p \mid \neg\phi \mid \phi \wedge \phi' \mid B_i\phi$$

where $p \in \mathcal{P}$, $i \in Ag$, $\phi \in \mathcal{L}$ and $B_i\phi$ means that *"agent i believes $\phi$."* The semantics for formulae in $\mathcal{L}$ is given by

Kripke models (Fagin et al. 2004) which are triplets, $M = \langle W, R, V \rangle$, containing a set of worlds, accessibility relations between the worlds for each of the agents ($R = \{R_i \mid i \in Ag\}$), and a valuation map, $V: W \to 2^P$. The set of worlds an agent $i$ (at world $w \in W$) considers possible is given by $M$ and the accessibility relations in $R_i$ pertaining to $w$. $R_i$ is a binary relation on $W$ and is a subset of $W \times W$. A formula $\phi$ is true in a world $w$ of a Kripke model $M = \langle W, R, V \rangle$, written $M, w \models \phi$, under these conditions:

$M, w \models p$ for an atom $p$, *iff* $p \in V(w)$,

$M, w \models \neg\phi$, *iff* $M, w \not\models \phi$,

$M, w \models \phi \wedge \psi$, *iff* both $M, w \models \phi$ and $M, w \models \psi$,

$M, w \models B_i\phi$, *iff* $M, w' \models \phi \quad \forall w' \in W$ s.t. $R_i(w, w')$

$\phi$ is satisfiable if there is a Kripke model $M$ and a world $w$ of $M$ s.t. $M, w \models \phi$. $\phi$ is said to entail $\psi$, written $\phi \models \psi$, if for any Kripke model $M$, $M, w \models \phi$ entails $M, w \models \psi$. We are interested in a set of properties of belief and assume a number of constraints on Kripke models to achieve this (Fagin et al. 2004). In particular, we assume that the Kripke model is:

*Serial* - $\forall w \, \exists v \, R(w, v)$

*Transitive* - $R(w, v) \wedge R(v, u) \Rightarrow R(w, u)$

*Euclidean* - $R(w, v) \wedge R(w, u) \Rightarrow R(v, u)$

with the resulting properties of belief:

$B_i\phi \wedge B_i(\phi \Rightarrow \psi) \Rightarrow B_i\psi$ (K - Distribution)

$B_i\phi \Rightarrow \neg B_i\neg\phi$ (D - Consistency)

$B_i\phi \Rightarrow B_iB_i\phi$ (4 - Positive Introspection)

$\neg B_i\phi \Rightarrow B_i\neg B_i\phi$ (5 - Negative Introspection)

We end up with the KD45$_n$ system (where $n$ is the number of agents in environment) that is defined by these properties of belief.

**Epistemic planning** combines automated planning – the task of selecting a goal-leading plan based on a description of the world – and reasoning over the beliefs and knowledge of agents. We appeal to a multi-agent epistemic planning formulation in this work to represent the beliefs of different agents in a dynamic setting.

**Definition 1** (MEP Problem). *A* **Multi-agent Epistemic Planning Problem** *is a tuple $\langle Q, \mathcal{I}, G \rangle$ where $Q = \langle \mathcal{P}, \mathcal{A}, Ag \rangle$ is the* **domain** *comprising sets of atoms $\mathcal{P}$, actions $\mathcal{A}$, and agents $Ag$, together with the problem instance description comprising the initial state, $\mathcal{I} \in \mathcal{L}$, and the goal condition $G \in \mathcal{L}$, where $\mathcal{L}$ is the language of multi-agent modal logic corresponding to $\mathcal{P}$ and $Ag$.*

States are formulae in the language – for example, for some atom $p \in \mathcal{P}$ and agents $i, j \in Ag$, $p$ is one state, $p \wedge B_ip$ is another state, and $p \wedge B_ip \wedge B_iB_jp$ is yet another state. $\mathcal{A}$ is a set of actions where each action $a \in \mathcal{A}$ is a tuple $\langle \text{PRE}, \{(\gamma_1, \epsilon_1), ..., (\gamma_k, \epsilon_k)\} \rangle$, where $\text{PRE} \in \mathcal{L}$ is the precondition of $a$ (written $\text{PRE}(a)$), $\gamma_i \in \mathcal{L}$ is the condition of a conditional effect, and $\epsilon_i \in \mathcal{L}$ is the effect of a conditional effect.

We assume, for any domain $Q = \langle \mathcal{P}, \mathcal{A}, Ag \rangle$ in question, the existence of a *progression* (Lin and Reiter 1997) operator $\text{PROG}(a, \phi)$, which given a formula $\phi$ (representing a state) and an action $a \in \mathcal{A}$, outputs another formula representing the changed state of the environment as the result of performing $a$. See (Lin and Reiter 1997) for how progression can be defined in STRIPS and the situation calculus. In this paper we rely on the definition and realization of progression by the epistemic planner used in Section 5.

### Notation

- We use the shorthand $\text{PROG}([a_1, ..., a_n], \phi)$ or $\text{PROG}(\pi, \phi)$ to denote the progression of $\phi$ wrt a sequence of actions $\pi = [a_1, ..., a_n]$.

- When talking about a domain $\langle \mathcal{P}, \mathcal{A}, Ag \rangle$, we will use $\mathcal{L}$ to refer to the multi-agent modal language corresponding to $\mathcal{P}$ and $Ag$.

- We use $\mathcal{I}$ and $S$ to denote states (formulas in $\mathcal{L}$), and use $G$ to denote a goal (also a formula in $\mathcal{L}$, which can encode an epistemic goal).

- When talking about a tuple $\langle v_1, \ldots, v_k \rangle$ of agents from $Ag$, we may use $\vec{B}$ to stand for the belief operator sequence $B_{v_1} \ldots B_{v_k}$.

## 3 Resolving Discrepancies

For a plan to achieve some goal – to be valid – a set of sufficient and necessary conditions must hold.

**Definition 2** (Plan Validity). *Given a domain $\langle \mathcal{P}, \mathcal{A}, Ag \rangle$, a state $S$, and a goal $G$, a plan $a_1, \ldots, a_k$ (where each $a_i \in \mathcal{A}$) is valid for achieving $G$ from $S$ if $\text{PROG}([a_1, \ldots, a_k], S) \models G$ and, for each $j < k$, $\text{PROG}([a_1, \ldots, a_j], S) \models \text{PRE}(a_{j+1})$.*

Suppose that we have (given a domain) a *regression* operator $\text{REG}$ (Reiter 2001) which maps a formula $\phi$ and action sequence $\pi$ to a formula $\text{REG}(\pi, \phi)$ which satisfies the property that for any state $S$, $S \models \text{REG}(\pi, \phi)$ just in case $\text{PROG}(\pi, S) \models \phi$.

**Definition 3** (VALID). *Let $\text{VALID}([a_1, \ldots, a_k], G)$ be an abbreviation for the formula*

$$\text{REG}([a_1, \ldots, a_k], G) \wedge \bigwedge_{j=0}^{k-1} \text{REG}([a_1, \ldots, a_j], \text{PRE}(a_{j+1})).$$

Observe that, given a domain $\langle \mathcal{P}, \mathcal{A}, Ag \rangle$, a plan $\pi$ to achieve $G$ is valid in $S$ just in case $S \models \text{VALID}(\pi, G)$. Since $\text{VALID}(\pi, G)$ is a formula, we can also talk about agents' beliefs about it, which we can interpret as indicating their beliefs about whether $\pi$ is a valid plan.

**Definition 4** (Subjective Plan Validity). *Given a domain $\langle \mathcal{P}, \mathcal{A}, Ag \rangle$, a state $S$, and a goal $G$, agent $i$ believes that a plan $\pi$ is valid if $S \models B_i\text{VALID}(\pi, G)$.*

**Observation 1.** *A plan may not be objectively valid but may be subjectively valid from some agent $i$'s perspective (and vice versa). That is, for some plan $\pi$ and goal $G$, it may be the case that in state $S$, $S \models \neg\text{VALID}(\pi, G) \wedge B_i\text{VALID}(\pi, G)$ (or $S \models \text{VALID}(\pi, G) \wedge B_i\neg\text{VALID}(\pi, G)$).*

Agents can also hold beliefs about other agents' beliefs (about other agents' beliefs) about the validity of a plan and perceive *discrepancies* between their beliefs and the beliefs of other agents about plan validity.

**Definition 5** (Discrepancy). *Given a domain* $\langle \mathcal{P}, \mathcal{A}, Ag \rangle$, *agents* $i, j \in Ag$, *and a (possibly empty) tuple* $\vec{v}$ *of agents in* $Ag$, *we say that a formula* $\phi$ *is a* **discrepancy** *perceived by agent* $i$ *in state* $S$ *between its beliefs and those of agent* $j$ *(about the beliefs of the agents in* $\vec{v}$, *if that tuple is non-empty) if the following condition is entailed by* $S$:

$$B_i \vec{B} \phi \wedge \neg B_i B_j \vec{B} \phi$$

*where* $\vec{B}$ *denotes the sequence of operators* $B_{v_1} \ldots B_{v_k}$, *where* $v_1, \ldots, v_k$ *are the elements of* $\vec{v}$.

Agent $i$ may also perceive discrepancies between its own (un)certainty and the (un)certainty of agent $j$ (e.g., $\neg B_i \vec{B} \phi \wedge B_i B_j \vec{B} \phi$). While here we do not address this setting, additional conditions can be straightforwardly added to Definition 5 to capture discrepancies of this nature.

We will be interested in discrepancies about formulas like $\text{VALID}(\pi, G)$, i.e., in discrepancies about the validity of plans, and of course in enabling agents to resolve such discrepancies by correcting other agents' beliefs. The following definition is of a plan that ensures that in the end, $\text{VALID}(\pi, G)$ (or $\neg \text{VALID}(\pi, G)$) will *not* be a discrepancy perceived by agent $i$ between its beliefs and those of agent $j$ (about the beliefs of the agents in $\vec{v}$).

**Definition 6** (Discrepancy Resolving Plan). *Given a domain* $Q = \langle \mathcal{P}, \mathcal{A}, Ag \rangle$, *agents* $i, j \in Ag$, *a (possibly empty) tuple* $\vec{v}$ *of agents in* $Ag$, *initial state* $\mathcal{I}$, *a plan* $\pi$, *and a goal* $G$, *a* **discrepancy resolving plan** *for* $\langle Q, \mathcal{I}, i, j, \vec{v}, \pi, G \rangle$ *is a plan* $\pi'$ *such that:*

1. *If* $\mathcal{I} \models B_i \vec{B} \text{VALID}(\pi, G)$ *then* $\mathcal{I} \models$ $\text{VALID}(\pi', B_i B_j \vec{B} \text{VALID}(\pi, G) \wedge B_i \vec{B} \text{VALID}(\pi, G))$.

2. *If* $\mathcal{I} \models B_i \vec{B} \neg \text{VALID}(\pi, G)$ *then* $\mathcal{I} \models$ $\text{VALID}(\pi', B_i B_j \vec{B} \neg \text{VALID}(\pi, G) \wedge B_i \vec{B} \neg \text{VALID}(\pi, G))$.

Agent $i$'s perceived discrepancy between its beliefs and the beliefs of agent $j$ is resolved via a *discrepancy resolving plan* $\pi'$ which is a sequence of communication and/or world altering actions. After $\pi'$'s execution, agent $i$ will believe that agent $j$ believes that (the agents in $\vec{v}$ believe that) the plan $\pi$ is valid (in case agent $i$ believes that (the agents in $\vec{v}$ believe that) $\pi$ is valid) or that agent $j$ believes that (the agents in $\vec{v}$ believe that) $\pi$ is not valid (in case agent $i$ believes that (the agents in $\vec{v}$ believe that) $\pi$ is not valid). Importantly, by including the conjunctive $B_i \vec{B} \text{VALID}(\pi, G)$ (resp. $B_i \vec{B} \neg \text{VALID}(\pi, G)$) in conditions (1) and (2), we ensure that executing the plan $\pi'$ in $\mathcal{I}$ does not change agent $i$'s own belief about whether $\text{VALID}(\pi, G)$ holds.

Finally, while in this work we focus on discrepancy resolving plans comprising explicit communicative actions (e.g., where agent $i$ communicates information to agent $j$), such plans may include different modes of communication (e.g., plans that may either *show* or *tell* another agent salient information to resolve discrepancies (Sreedharan et al. 2020a)).

**Example.** Consider a search and rescue scenario with three agents, Alice, Bob, and Mary, where all agents are aware that Bob's goal is to obtain a particular medical kit (Med-Kit1). The search and rescue setting is discussed in more detail in Section 5. Initially, Alice believes that Bob falsely believes that MedKit1 is in room A (Alice herself believes that the medical kit is in room B). Alice also believes that Mary falsely believes that Bob believes that MedKit1 is in room B. All agents believe that in order to pick up a medical kit, one must be in the same room as the medical kit. We partially model this scenario:

$$Ag = \{\text{Alice, Mary, Bob}\} \tag{1}$$
$$\mathcal{I} \models B_{\text{Alice}} at(\text{Bob, HallWay}) \tag{2}$$
$$\mathcal{I} \models B_{\text{Alice}} at(\text{MedKit1, RoomB}) \tag{3}$$
$$\mathcal{I} \models B_{\text{Alice}} \neg at(\text{MedKit1, RoomA}) \tag{4}$$
$$\mathcal{I} \models B_{\text{Alice}} B_{\text{Mary}} at(\text{MedKit1, RoomB}) \tag{5}$$
$$\mathcal{I} \models B_{\text{Alice}} B_{\text{Bob}} at(\text{MedKit1, RoomA}) \tag{6}$$
$$\mathcal{I} \models B_{\text{Alice}} B_{\text{Bob}} \neg at(\text{MedKit1, RoomB}) \tag{7}$$
$$\mathcal{I} \models B_{\text{Alice}} B_{\text{Mary}} B_{\text{Bob}} at(\text{MedKit1,RoomB}) \tag{8}$$
$$\mathcal{I} \models B_{\text{Alice}} B_{\text{Mary}} B_{\text{Bob}} \neg at(\text{MedKit1,RoomA}) \tag{9}$$

In Section 4 we discuss computational techniques with which Alice can predict other agents' plans (as well as other agents' predictions about other agents' plans). For now, let us assume that the goal $G$ is *holding*(Bob,MedKit1) and that Alice predicts that Bob's plan to achieve $G$ is

[*move*(Bob,HallWay,RoomA), *pickUp*(Bob,MedKit1,RoomA)].

We refer to Alice's prediction about Bob's plan as $\pi_{\text{AliceBob}}$. The actions in $\pi_{\text{AliceBob}}$ are modelled as follows:

*move*(Bob,HallWay,RoomA) = $\langle at(\text{Bob, HallWay}),$ $\{(\top, at(\text{Bob, RoomA})), (\top, \neg at(\text{Bob, HallWay}))\}\rangle$

*pickUp*(Bob,MedKit1,RoomA) = $\langle at(\text{MedKit1, RoomA}),$ $\{(\top, holding(\text{Bob, MedKit1})), (\top, \neg at(\text{MedKit1, RoomA}))\}\rangle$

And so we have that

$$\text{VALID}(\pi_{\text{AliceBob}}, G) =$$
$$at(\text{MedKit1, RoomA}) \wedge at(\text{Bob, HallWay})$$

Moreover, Alice can reason that Mary predicts that Bob's plan is

[*move*(Bob,HallWay,RoomB),*pickUp*(Bob,MedKit1,RoomB)].

We refer to Alice's prediction about Mary's prediction about Bob's plan as $\pi_{\text{AliceMaryBob}}$. Given entailments (2)-(9), the following holds pertaining to agents' beliefs about the validity of $\pi_{\text{AliceBob}}$ and $\pi_{\text{AliceMaryBob}}$:

$$\mathcal{I} \models B_{\text{Alice}} \neg \text{VALID}(\pi_{\text{AliceBob}}, G) \tag{10}$$
$$\mathcal{I} \models B_{\text{Alice}} B_{\text{Bob}} \text{VALID}(\pi_{\text{AliceBob}}, G) \tag{11}$$
$$\mathcal{I} \models B_{\text{Alice}} B_{\text{Mary}} B_{\text{Bob}} \neg \text{VALID}(\pi_{\text{AliceBob}}, G) \tag{12}$$
$$\mathcal{I} \models B_{\text{Alice}} \text{VALID}(\pi_{\text{AliceMaryBob}}, G) \tag{13}$$
$$\mathcal{I} \models B_{\text{Alice}} B_{\text{Bob}} \neg \text{VALID}(\pi_{\text{AliceMaryBob}}, G) \tag{14}$$
$$\mathcal{I} \models B_{\text{Alice}} B_{\text{Mary}} B_{\text{Bob}} \text{VALID}(\pi_{\text{AliceMaryBob}}, G) \tag{15}$$

Alice perceives in $\mathcal{I}$ a number of discrepancies between her beliefs and those of Bob and Mary pertaining to plan validity. In particular, $\text{VALID}(\pi_{\text{AliceBob}}, G)$ is a discrepancy perceived by Alice between her beliefs and those of Bob, where $\vec{v}$ is empty (entailments (10) and (11)). Since $\mathcal{I} \models B_{\text{Alice}}\neg\text{VALID}(\pi_{\text{AliceBob}}, G)$, a discrepancy resolving plan for Alice's perceived discrepancy is

$$\pi' = [inform(\text{Alice},\text{Bob},\neg at(\text{MedKit1},\text{RoomA}))]$$

where

$$\mathcal{I} \models \text{VALID}(\pi', B_{\text{Alice}}B_{\text{Bob}}\neg\text{VALID}(\pi_{\text{AliceBob}}, G))$$

The inform action in $\pi'$ is modelled as follows:

$$inform(\text{Alice},\text{Bob},\neg at(\text{MedKit1},\text{RoomA})) =$$
$$\langle B_{\text{Alice}}\neg at(\text{MedKit1, RoomA}),$$
$$\{(\top, B_{\text{Alice}}B_{\text{Bob}}\neg at(\text{MedKit1, RoomA}))\}\rangle$$

The discrepancy resolving plan $\pi'$ consists of Alice informing Bob that MedKit1 in not in room A. This resolves Alice's perceived discrepancy about the validity of $\pi_{\text{AliceBob}}$. That is, Alice believes that after Bob learns that MedKit1 is not in room A, he will believe that the plan $\pi_{\text{AliceBob}}$ is not valid. In Section 4 we discuss how to leverage epistemic planning to compute discrepancy resolving plans.

Note that modelling the inform action in this way enforces truthful communication, since a precondition of this action is that Alice believe $\neg at(\text{MedKit1, RoomA})$. Moreover, Alice believing that Bob will revise his beliefs appropriately following the inform action is predicated on Alice believing that Bob will find her communication trustworthy. An interesting avenue for future work is the integration of trust into our framework (see, for example, Fabiano's (2020) discussion of trust in epistemic planning).

There is also a 'higher-order' discrepancy in our example. In particular, $\text{VALID}(\pi_{\text{AliceBob}}, G)$ is a discrepancy perceived by Alice between her beliefs and those of Mary about Bob's beliefs, where $\vec{v}$ is $\langle\text{Bob}\rangle$. That is, while Alice believes that Bob believes that $\pi_{\text{AliceBob}}$ is valid (entailment (11)), she also believes that Mary believes that Bob believes that $\pi_{\text{AliceBob}}$ is not valid (entailment (12)). This is because of Mary's false belief about Bob's belief about the medical kit's location (entailments (8) and (9)). Since $\mathcal{I} \models B_{\text{Alice}}B_{\text{Bob}}\text{VALID}(\pi_{\text{AliceBob}}, G)$, a discrepancy resolving plan in this case is

$$\pi' = [inform(\text{Alice},\text{Mary},B_{\text{Bob}}at(\text{MedKit1},\text{RoomA}))]$$

such that

$$\mathcal{I} \models \text{VALID}(\pi', B_{\text{Alice}}B_{\text{Mary}}B_{\text{Bob}}\text{VALID}(\pi_{\text{AliceBob}}, G))$$

Alice believes that after Mary learns that Bob believes that MedKit1 is in room A, Mary will believe that Bob believes that $\pi_{\text{AliceBob}}$ is valid (which resolves the perceived discrepancy).

## 4 Computing Discrepancy Resolving Plans

As mentioned, epistemic planning combines automated planning and reasoning over the beliefs and knowledge of agents. In this section we discuss a method with which to compute discrepancy resolving plans using epistemic planning tools, and also discuss how an agent can detect discrepancies relating to plan validity.

Algorithm 1 accepts as input a tuple $R = \langle Q, \mathcal{I}, i, j, \vec{v}, \pi, G\rangle$ and returns a discrepancy resolving plan for it. In Line 3, the formula $\phi = \text{VALID}(\pi, G)$ is computed given $R$, using an implementation of the regression operator REG (the COMPUTEREGRESSIONFORMULA function). Subsequently, an epistemic planner may be tasked (the CALLEPISTEMICPLANNER function) with solving one of

$$\langle Q, \mathcal{I}, B_i B_j \vec{B}\phi \wedge B_i \vec{B}\phi\rangle$$

and

$$\langle Q, \mathcal{I}, B_i B_j \vec{B}\neg\phi \wedge B_i \vec{B}\neg\phi\rangle$$

depending on whether $\mathcal{I} \models B_i\vec{B}\phi$ or $\mathcal{I} \models B_i\vec{B}\neg\phi$. The epistemic goal given to the planner ensures that the solution returned by the planner, $\pi'$, is a discrepancy resolving plan.

---

**Algorithm 1**

1: **procedure** RESOLVEDISCREPANCY($\langle Q, \mathcal{I}, i, j, \vec{v}, \pi, G\rangle$)
  Given a tuple $R = \langle Q, \mathcal{I}, i, j, \vec{v}, \pi, G\rangle$, return a discrepancy resolving plan.
2:     $\pi' \leftarrow []$
3:     $\phi \leftarrow$ COMPUTEREGRESSIONFORMULA($R$)
4:     **if** $\mathcal{I} \models B_i\vec{B}\phi$ **then**
5:       $G' \leftarrow B_i B_j \vec{B}\phi \wedge B_i\vec{B}\phi$
6:       $\pi' \leftarrow$ CALLEPISTEMICPLANNER($\langle Q, \mathcal{I}, G'\rangle$)
7:     **end if**
8:     **if** $\mathcal{I} \models B_i\vec{B}\neg\phi$ **then**
9:       $G' \leftarrow B_i B_j \vec{B}\neg\phi \wedge B_i\vec{B}\neg\phi$
10:      $\pi' \leftarrow$ CALLEPISTEMICPLANNER($\langle Q, \mathcal{I}, G'\rangle$)
11:     **end if**
12:     **return** $\pi'$
13: **end procedure**

---

### 4.1 Detecting Discrepancies

An agent may leverage Algorithm 1 to compute discrepancy resolving plans. Importantly, discrepancy resolving plans should be computed when there is need for them, i.e., when an agent detects that there are discrepancies between her beliefs and the beliefs of other agents about the validity of some plan. One way for an agent to detect such discrepancies is by *predicting* how other agents (predict that other agents) plan to achieve some goal. More generally, this prediction task is akin to the *plan recognition* task where an observing agent attempts to predict an observed agent's plan and goal given a sequence of observations about the world and the behavior of the observed agent (e.g., Kautz 1987 ; Ramírez and Geffner 2010). Here we are interested in a simple version of this problem, wherein the goal is known and no observations are provided. Shvo et al. (2020) have shown how epistemic planning tools can be used to perform

plan recognition in the context of agent beliefs and epistemic goals. Future work can naturally extend the approach presented here using Shvo et al.'s framework and address settings where an observer attempts to detect discrepancies between its beliefs and the beliefs of the observed agent(s).

We define a function PREDICTPLAN($\mathcal{P},\mathcal{A},Ag,\mathcal{I},G,\vec{v}$) where $\langle \mathcal{P}, \mathcal{A}, Ag \rangle$ is a domain, $\mathcal{I}$ is the initial state, $G$ is a goal, and $\vec{v}$ is a tuple of agents. PREDICTPLAN returns a plan $\pi$ such that

$$\mathcal{I} \models \vec{B}\text{VALID}(\pi, G)$$

where $\vec{B}$ corresponds to the (possibly empty) tuple of agents $\vec{v}$. For example, if $\vec{v} = \langle i, j, k \rangle$, $\pi$ is agent $i$'s prediction about agent $j$'s prediction about agent $k$'s plan to achieve $G$ (note that agent $i$ need not believe that $\pi$ is valid). Here we realize this function by tasking an epistemic planner with solving a MEP problem $\langle \langle \mathcal{P}, \mathcal{A}, Ag \rangle, \mathcal{I}, G' \rangle$ where $G' = B_i B_j B_k G$. Note that there may be multiple optimal plans that solve the MEP problem and the planner is assumed to return one at random. Alternatively, future work could draw on the probabilistic plan recognition literature, where a common solution to an agent's uncertainty about the plans of other agents is to compute a probability distribution over the set of possible plans.

Given some goal $G$, agent $i$ can use the function PREDICTPLAN to predict the plans of each agent in $Ag$ and the predictions of each agent in $Ag$ about the plans of all other agents and so on. If agent $i$ perceives a discrepancy between its beliefs and those of other agents about the validity of any of the predicted plans, Algorithm 1 can be used to resolve it.

**Example.** In our example, Bob has the goal of obtaining a medical kit. When PREDICTPLAN is called with that goal and the tuple of agents $\langle$Alice,Bob$\rangle$, it returns the plan $\pi_{\text{AliceBob}}$, i.e.,

[*move*(Bob,HallWay,RoomA), *pickUp*(Bob,MedKit1,RoomA)].

While Alice believes that $\pi_{\text{AliceBob}}$ is not valid, she believes that Bob believes it is valid (entailments (9) and (10)). Since Alice perceives this as a discrepancy, Algorithm 1 may be called to compute an appropriate discrepancy resolving plan. Moreover, when PREDICTPLAN is called with Bob's goal and the tuple of agents $\langle$Alice,Mary,Bob$\rangle$, it returns the plan $\pi_{\text{AliceMaryBob}}$, i.e.,

[*move*(Bob,HallWay,RoomB),*pickUp*(Bob,MedKit1,RoomB)].

In this case, while Alice believes that Mary believes that $\pi_{\text{AliceMaryBob}}$ is valid, Alice also believes that Bob believes it is not valid (entailments (14) and (15)). As before, Algorithm 1 may be called.

# 5 Evaluation

In this section, we present the results of our evaluation, where we set out to demonstrate that existing epistemic planners can be used to compute discrepancy resolving plans in a number of domains. We also report the results of a study we conducted to evaluate our approach's ability to resolve participants' misconceptions regarding plan validity.

We make use of the latest version of RP-MEP[1], an off-the-shelf epistemic planner (Muise et al. 2015). Muise et al.'s planner encodes a multi-agent epistemic planning problem as a classical planning problem and augments actions in the domain with special conditional effects that enforce the KD45 axioms. Muise et al.'s MEP formulation uses a syntactically restricted fragment of epistemic logic to mitigate for the computational complexity of reasoning in multi-agent epistemic logic. To this end, i. reasoning in Muise et al.'s framework is done from the perspective of a single root agent, ii. an upper bound is set on the depth of nested reasoning, and iii. disjunctive belief is not allowed. The classically encoded MEP problem can then be given to an off-the-shelf classical planner. To run Algorithm 1, RP-MEP was called in Lines 6 and 10 with the Fast Downward planner (Helmert 2006) with an admissible heuristic. Our implementation of the regression operator (used in Line 3 of Algorithm 1) is given as input the classically encoded MEP problem.

In what follows, we describe the various domains (and problems within those domains) used in our experiments. All problem instances across all domains were modelled as tuples comprising a domain, an initial state, two agents, a (possibly empty) tuple of agents, a plan, and a goal, and given to Algorithm 1. In our experiments we did not evaluate agents' ability to predict other agents' plans. Instead, all plans in the tuples given to Algorithm 1 were pre-computed using the PREDICTPLAN function described in Section 4.1. PREDICTPLAN was realized using RP-MEP.

**Observability in domains in evaluation:** All ontic actions (e.g., movement, picking up and dropping off an object) are observable by all agents in the room/location in which the action is performed. Therefore, agents' belief about some object will not change if they are not in the room when that object is picked up and taken out of the room. To enable this, we make use of the *conditioned mutual awareness* mechanism implemented in the RP-MEP planner (for more details, see (Muise et al. 2015)).

**BW4T** Johnson et al. (2009) presented a multi-agent simulation platform, BlocksWorld for Teams (BW4T), which is an abstraction of a myriad of application domains such as search & rescue. Typically in this domain, there are a number of rooms and a drop zone, where each room contains a number of colored blocks. In the application domains, blocks may represent survivors of a disaster or medical kits, and the various agents may be humans or robots with different roles and capabilities. We cast blocks as medical kits.

We modelled[2] various instances of the BW4T domain by varying the number of rooms, medical kits, and types of medical kits, totalling 5 unique problem instances. Moreover, in each instance we modelled a number of scenarios involving perceived discrepancies about plan validity. Common to all scenarios and instances is the following: there are three agents in the environment (Alice, Mary, and Bob); all discrepancies are perceived by Alice and resolved by her; and all plans (except for the discrepancy resolving plans) are

---

[1]https://github.com/QuMuLab/pdkb-planning

[2]by modifying the domains in https://bit.ly/3lWjJ3b

executed by Bob. In all scenarios, Alice can (truthfully) inform other agents of either (agents' beliefs about) the whereabouts of various medical kits or the status of Mary's communication device.

*Dude, Where's my Medical Kit?* This scenario (and the next) builds on our running example. Bob has as his goal to get a particular medical kit to the drop zone and believes that the medical kit is in some room. Alice believes that Bob holds a false belief pertaining to the location of the medical kit. Two discrepancy resolving plans are computed: one to explain to Bob that his plan ($\pi_{\text{AliceBob}}$ in our example in Section 3) is not valid and the other to explain to Bob that the plan $\pi_{\text{AliceMaryBob}}$ is valid (from Alice's perspective, based on her belief about the location of the medical kit). $\vec{v}$ is empty.

*Where Does he Think he's Going?* In this scenario, the setup is the same but Mary is involved and Alice believes that Mary falsely believes that Bob does not have a false belief about the location of the medical kit (similarly to our running example). Two discrepancy resolving plans are computed: one to explain to Mary that Bob does not believe that $\pi_{\text{AliceMaryBob}}$ is valid and another to explain to Mary that Bob believes that $\pi_{\text{AliceBob}}$ is valid. $\vec{v}$ is $\langle \text{Bob} \rangle$.

*Can You Hear Me??* Here, Bob has the goal of getting a particular medical kit to the drop zone and notifying his teammate, Mary, that he has done so. This time, Bob has a correct belief about the medical kit's location. However, while Alice believes that Mary's communication device is not working properly (perhaps she met Mary in passing and was told by her), she also believes that Bob falsely believes that it is working properly. Alice therefore believes that Bob's plan (which involves sending a message to Mary) will fail to achieve the *epistemic* component of his goal (i.e., for Mary to believe that a medical kit is now at the drop zone). A discrepancy resolving plan is computed where $\vec{v}$ is empty.

**Epistemic Planning Benchmarks** In our modified[3] version of the *Corridor* domain, there are $n$ agents in various rooms connected to a long corridor. A single acting agent, Bob, holds a secret and can move along the corridor, enter different rooms, and announce his secret. When announcing the secret, all agents in the room with the announcer, as well as (due to extremely thin walls) all agents in the adjacent rooms, now believe the secret. Bob may have different epistemic goals, including a universal or selective spread of his secret to the other agents. We create 5 instances of this domain by varying the number of rooms, agents, and the false beliefs held by Bob about the locations of each agent. For each of the generated instances, a discrepancy resolving plan is computed where $\vec{v}$ is empty and the agent perceiving the discrepancy is Alice (an agent in the environment who holds correct beliefs about agent locations and about Bob's beliefs). Alice can inform Bob of agents' locations.

There are two reasons for Alice to believe that Bob's plan is not valid (while believing that he believes it is valid): i. Bob's goal is for agent $i$ to believe his secret but Alice believes that Bob falsely believes that $i$ is in some room and will plan to head to that room to share his secret with $i$; ii.

---

[3]The unmodified corridor domain can be found in https://bit.ly/3tXsi0r.

Bob only wants agent $i$ to believe his secret without agent $j$ believing it. However, Bob falsely believes that agent $j$ is neither in the room with agent $i$ nor in the adjacent rooms. Therefore, Bob will plan to go to the room in which he believes $i$ to be and share his secret with her. However, this plan will fail to achieve his goal since agent $j$, who is in the adjacent room, will also come to believe Bob's secret.

**IPC Domains** Inspired by 7 IPC domains[4], we modelled 7 MEP domains with agents Alice, Bob, and Mary. We generated 35 problem instances (5 from each domain) by creating false beliefs for agents (e.g., causing an agent to hold a false belief about the location of an object in the Driverlog domain) and varying the domain parameters (e.g., number of objects in the domain). For each problem instance, we experimented with two types of discrepancy resolutions: where Alice (who perceives all discrepancies) explains to Bob why his plan is (in)valid and where Alice explains to Mary why Bob believes his plan is (in)valid. In each domain, Alice has at her avail appropriate communicative actions.

**Results** Table 1 summarizes the results for the various domains. The $T$ values are the average runtime (in seconds) for Algorithm 1 (using RP-MEP) over 5 problem instances of the respective domain. $d$ is the required depth of nested belief, which is 2 for problems where $\vec{v}$ is empty (and non-epistemic goals) and 3 for problems where $\vec{v}$ comprises 1 agent (as well as problems with epistemic goals). The variances ranged 0.07-0.45 for the T values. Finally, all discrepancy resolving plans consisted of 1-3 inform actions.

Table 1 also shows that $d$, the depth of nested belief, impacts RP-MEP's runtime. This is because the number of new fluents introduced during RP-MEP's encoding process is exponential in $d$. Le et al. (2018) and Shvo et al. (2020) empirically show that there exist other epistemic planners whose performance is not affected by the value of $d$. In settings where high depth of reasoning is required, the application of these planners to discrepancy resolution could be investigated.

## 5.1 Usability Study

So far in this section we have demonstrated that existing epistemic planners can be used to compute discrepancy resolving plans in a number of domains. However, these results are not necessarily a testament to the efficacy of our approach in the *presence of humans*. As such, we conducted a usability study aimed to evaluate the ability of our approach, which is described in Sections 3 and 4, to resolve participants' misconceptions regarding plan validity. Specifically, we set out to test the following hypotheses:

**H1:** Participants will be more likely to generate a valid plan to achieve their goal when presented with information derived from a discrepancy resolving plan, compared to the likelihood of generating a valid plan prior to receiving the information.

**H2:** Participants will more accurately predict another agent's plan when presented with information derived

---

[4]Which can be found in http://editor.planning.domains/

| Domain | d | T (Alg 1) | Domain | d | T (Alg 1) | Domain | d | T (Alg 1) |
|---|---|---|---|---|---|---|---|---|
| BW4T | 2 | 1.93 | Driverlog | 2 | 1.51 | Logistics | 2 | 41.84 |
| BW4T | 3 | 2.31 | Driverlog | 3 | 3.77 | Logistics | 3 | 59.33 |
| BW4T (EG) | 3 | 3.16 | Gripper | 2 | 1.33 | Zeno | 2 | 24.16 |
| Corridor (EG) | 3 | 2.94 | Gripper | 3 | 4.47 | Zeno | 3 | 42.69 |
| Depots | 2 | 20.93 | Rovers | 2 | 1.39 | Satellite | 2 | 2.38 |
| Depots | 3 | 37.41 | Rovers | 3 | 2.88 | Satellite | 3 | 4.25 |

Table 1: Average runtime in seconds (T) for Algorithm 1 and Algorithm 2 using RP-MEP. $d$ is the required depth of nested belief and EG signifies that problems in the domain involve an epistemic goal.

from a discrepancy resolving plan, compared to their prediction prior to receiving the information.

To test these hypotheses, participants were asked to imagine themselves as medics in a search & rescue team where they are partnered with a virtual assistant meant to provide decision support. Participants were presented with two scenarios, mirroring two of the BW4T scenarios discussed earlier in this section. We had a total of 36 participants who were recruited via Amazon Mechanical Turk and were paid upon completing the questionnaire via an online platform[5]. Participants had no prior knowledge about the study.

**Testing H1** In the first scenario (mirroring the *'Dude, Where's my Medical Kit?'* scenario), participants were told that their goal is to acquire a medical kit from the supply tent of the base. Subsequently, participants were given *incorrect* information about the location of the supply tent. Participants were then asked[6] where they would go in order to obtain a medical kit.

Next, participants were informed by their virtual assistant of the true location of the supply tent and were asked, again, where they would go in order to obtain a medical kit. The assistant's communication is a natural language representation of the inform actions in the discrepancy resolving plan generated by RP-MEP. That is, the content of an inform action is processed using a number of simple natural language templates. For example, $at$(SupplyTent,WestEnd) is converted to *"The supply tent is at the west end"*. The discrepancy resolving plan resolves a discrepancy perceived by the virtual assistant (who is assumed to hold correct beliefs) between its beliefs and its beliefs about the participant's beliefs, pertaining to the validity of participants' plans.

**Testing H1 - Results** Prior to being informed by the virtual assistant about the true location of the supply tent, 0 participants generated a valid plan to obtain the medical kit. After being informed by the virtual assistant about the true location of the medical kit, 97% of participants correctly generated a valid plan to obtain the medical kit, which indicates that their misconception regarding plan validity was resolved.

Results comparing the likelihood of participants generating a valid plan to achieve their goal demonstrated a statistically significant difference between participants' predic-

[5]https://www.surveymonkey.com/
[6]Participants were given 4 options from which to choose: the west, east, north, and south ends of the base.

tions before receiving information from the virtual assistant ($M$=0, $SD$=0), and after receiving information from the virtual assistant ($M$=0.97, $SD$=0.17), $t$(35)= 35, $p < 0.001$.

These results are consistent with **H1** since participants were more likely to generate a valid plan to achieve their goal *after* being presented with information derived from a discrepancy resolving plan, compared to the likelihood of generating a valid plan.

**Testing H2** In the second scenario (mirroring the *'Where Does he Think he's Going?'* scenario), participants were told that they have a human teammate whose goal it is to acquire a medical kit from the supply tent. Participants were led to *incorrectly* believe that their teammate has been informed of the new location of the supply tent and were not aware that their teammate, in fact, has not received this information because of the teammate's faulty communication device. Participants were then asked to predict the location to which their teammate will go in order to obtain the medical kit. This is a proxy for participants' beliefs about their teammate's beliefs about plan validity. In other words, if participants believe that their teammate believes that one of the 4 possible plans is valid (i.e., going to a certain location) then we assume that participants will predict that their teammate will follow that plan and head to the respective location.

Next, participants were informed by their virtual assistant of their teammate's false belief and were asked, again, to predict the teammate's plan. As before, the assistant's communication is a natural language representation of the inform actions in the discrepancy resolving plan generated by RP-MEP. This time, the communication pertains to the teammate's beliefs. For example, $B_{\text{Mary}}at$(SupplyTent,WestEnd) is converted to *"Mary is not aware that the supply tent is at the west end"*. The discrepancy resolving plan resolves a discrepancy perceived by the virtual assistant (who is assumed to hold correct beliefs) between its beliefs and its beliefs about the participant's beliefs, pertaining to the validity of the teammate's plan.

**Testing H2 - Results** Prior to being informed by the virtual assistant about their teammate's beliefs, 0 participants correctly predicted their teammate's plan. After being informed by the virtual assistant about their teammate's beliefs, 80% of participants correctly predicted their teammate's plan which indicates that their misconception regarding plan validity was resolved.

Results comparing the accuracy of participants' predic-

tions about their teammate's plan demonstrated a statistically significant difference between participants' predictions before receiving information from the virtual assistant ($M$=0, $SD$=0), and after receiving information from the virtual assistant ($M$=0.72, $SD$=0.45), $t(35)$= 9.54, $p < 0.001$.

These results are consistent with **H2** since participants more accurately predicted their teammate's plan *after* being presented with information derived from a discrepancy resolving plan, compared to their prediction before receiving the information.

## 6  Related Work

There is a rich body of work related to the ideas discussed in this paper. In Explainable AI Planning (XAIP) (Chakraborti, Sreedharan, and Kambhampati 2020), a planning agent is tasked with explaining some aspect of plan generation or execution (e.g., optimality or validity). XAIP is a special case of the general task of explanation generation (e.g., Miller, 2019) which necessitates Theory of Mind reasoning in order to generate explanations that are meaningful to their recipient. Indeed, within XAIP, the body of work on *model reconciliation* has extensively investigated how to enable agents to use their Theory of Mind and resolve discrepancies between agents in the context of plan explanation (Chakraborti et al. 2017; Sreedharan, Chakraborti, and Kambhampati 2021). While most work in model reconciliation (and XAIP more broadly) has focused on enabling a planning agent to resolve discrepancies it perceives between its beliefs and those of a human observer about the agent's plan, model reconciliation techniques have also been used to resolve misconceptions held by humans about their plans (e.g., explaining to a human decision maker why her plan is not valid) with applications in proactive decision support systems (Grover et al. 2020) and dialogue modelling (Sreedharan et al. 2020b).

Within the body of work on model reconciliation, Sreedharan et al. (2020a) propose to leverage a simplified compilation to classical planning which draws from epistemic planning techniques (the original compilation is at the heart of the RP-MEP planner used in this work (Muise et al. 2015)). Our motivation and Sreedharan et al.'s overlap in that we are also interested in leveraging automated planning tools to resolve discrepancies between the mental models of agents. However, our work goes beyond the extant literature by broadening the discussion of the role of Theory of Mind in plan explanation. Namely, by appealing to an epistemic logic framework and by leveraging epistemic planners, we can model settings that previous work cannot, where agents can reason about the *nested beliefs* of other agents and resolve 'higher-order' discrepancies regarding the validity of plans, as well as correct misconceptions pertaining to the validity of plans pursuant to *epistemic goals*.

## 7  Concluding Remarks

In this work, we discussed how agents can use their Theory of Mind to resolve discrepancies between their beliefs and the beliefs of other agents regarding plan validity. Our proposed formulation appeals to epistemic logic and allows agents to reason about the nested beliefs of other agents and identify and repair beliefs that give rise to plan validity discrepancies. We realized our approach using epistemic planning and show how an epistemic planner may be used to generate discrepancy resolving plans in a number of domains. A study showcased our approach's ability to resolve misconceptions regarding plan validity held by humans.

Avenues for future work abound. For instance, we wish to explore synergies with the body of work on implicit coordination (Engesser et al. 2017; Engesser and Miller 2020), experiment with other epistemic planners, and relax the assumptions made in the realization of our approach. Additionally, in some cases (e.g., where two agents may need to hear the same piece of information), aggregating all pairwise communications (by running Algorithm 1 for all agents) might not be a desirable solution. Our definitions and algorithms can be straightforwardly adapted such that the epistemic goals given to the planner encourage a global optimization, rather than a pairwise one. Finally, future work will investigate how agents may resolve discrepancies regarding various plan attributes such as optimality.

**Acknowledgements:**  We gratefully acknowledge funding from NSERC, the CIFAR AI Chairs program, and Microsoft Research. We also acknowledge the rich multi-disciplinary research environment at the Schwartz Reisman Institute. Resources used in preparing this research were provided, in part, by the Province of Ontario, the Government of Canada through CIFAR, and companies sponsoring the Vector Institute for Artificial Intelligence (www.vectorinstitute.ai/partners).

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
