# OpenReview forum: "Explaining the Plans of Agents via Theory of Mind"
_icaps-conference.org/ICAPS/2021/Workshop/XAIP — XAIP 2021_

### Official Review · AnonReviewer1 · 2021-07-05
**Very interesting paper with some minor presentation issues**

**Rating:** 8
**Confidence:** 3

**Review:**

The authors present a way to reconcile the belief bases of agents w.r.t. their understanding of other agents' plans.
They do so via a framework rooted in epistemic logic.
To the best of my understanding, the authors build on existing results on epistemic planners showing how a reconciling plan can be automatically computed, provided that a discrepancy in agents' theories of others agents' minds can be identified.
In particular, the authors focus on the identification of such discrepancies.

The paper is clearly written, and well organized.
To the best of my knowledge, the paper contribution seems sound, accurate, and on-topic w.r.t. ICAPS.
The provided literature seems adequate and up to date.

The authors also describe a number of experiments based on f RP-MEP, an off-the-shelf epistemic planner written in Python.
If I well understood the structure of the code, the experiments presented by the authors are included among the examples available on the GitHub page of RP-MEP.

I only have a number of minor concerns about this paper:
1. Reading the paper is sometimes challenging because of the many formulae coming with poor informal intuitive descriptions. Consider for instance the background paragraphs about KD45n: that's essentially useless as the expert reader will find it very concise and skippable, whereas the inexpert reader will find it hard to read.
2. In algorithms 1 and 2 a number of sub-routines are mentioned, eg. TransformToMEP, ComputeRegressionFormula, etc.
I had to read the paper several times to figure out which exact formulae from sec 4 those sub-routines refer to... maybe explicitly referencing formulae via numbers may be useful here
3. Why do the authors present 2 algorithms doing the same thing in sec. 4? Isn't algorithm 2 better? What's the purpose of presenting paper 1 as well?
4. The author mentions a number of experiments based  RP-MEP, yet only the source code of RP-MEP is referenced.
It seems to be that the code of the experiments is part of the examples provided on the RP-MEP on GitHub.
If this is the case, I suggest separating the code of RP-MEP from the code of the experiments, e.g. via different repositories, providing detailed instructions on how to reproduce the experiments. If this is not the case, I suggest the author to make the code of their experiments publicly available for this paper to be considered for publication.

---

### Official Review · AnonReviewer2 · 2021-07-07
**Novel compilation method to generate discrepancy resolving plans via epistemic planning**

**Rating:** 8
**Confidence:** 4

**Review:**

The paper utilizes epistemic logic framework to generate a sequence of communication actions to resolve discrepancies between an agent’s belief and the belief of other agents regarding the validity of a query plan. It formally defines discrepancy of a plan validity with epistemic logic, then presents a compilation method to generate discrepancy resolving actions by solving a transformed epistemic planning problem (similar spirit of what Ramirez and Geffner (2009) did for solving plan recognition as classical planning). The paper describes the compilation, two solution methods with access to off-the-shelf epistemic planner and PREDICTPLAN function, and validation studies with experiments on benchmark domains and user studies.

Overall, the paper is written well, it is highly relevant to XAIP, and the methodology is sound. I think it’s a valuable contribution to the workshop. I highlight some of my key recommendations below.

I think the paper should discuss any related work on global optimization strategies of getting all agents on the same page in contrast to the pairwise framework adopted here. Ultimately on a team task, what would the agents each need to do to achieve global consensus? Simple aggregation of all pairwise communications wouldn’t be a desirable solution and may even cause interference.


The meaning and implication of an optimal solution to the transformed epistemic planning problem could have been expanded upon. Does the optimal solution here minimize the number of necessary communication actions? How important is it to have an optimal solution? What guarantees does the optimal solution provide for the problem?


The user studies could have been improved by comparing against baseline communication actions rather than to the extreme of not providing any communication. On simple benchmark problems, a randomly selected belief (assuming trustworthy which the paper adopts) or a heuristic like an agent’s belief about a goal predicate or something close to the goal, could perform very well and always help the listener (with extra information). Such more precise comparison could have clearly demonstrated the intelligent selection derived from the solutions to the presented approach.

---

### Meta-Review · Area_Chairs · 2021-07-07

**Recommendation:** Accept
**Confidence:** 5

**Metareview:**


Thanks for your submission to the workshop!

Summary: This paper proposes an approach for selecting communication actions that can resolve discrepancies between agents and their beliefs about the plans of agents and their validity. The reviewers agree that the paper is well written, that the topic of the presented work is interesting and it is relevant to the workshop.

Strengths:
* A clearly presented paper with an interesting and novel idea;
* The tackled problem is challenging, the presented framework provides a useful starting place and the paper includes useful discussion of next steps.

Limitations:
* The motivation for the type of reconciliation provided by the approach could be more clearly stated. Does the pairwise approach fit into a more general strategy? Or can the techniques presented build towards establishing sufficient knowledge for all agents?
* The reviewers identified certain presentational aspects (e.g., clearer labelling of algorithm functions, lack of intuition around definitions);
* The approach relies on knowing or guessing the other agent plans and the implications of this are not made clear.

Overall, an interesting work that can be a valuable contribution to the workshop. We hope that you find the reviewers' comments useful and that they will help you to revise your paper.

---

### Decision · Program_Chairs · 2021-07-08

Accept